# Metagenomic Next-Generation Sequencing Reveal Presence of a Novel Ungulate *Bocaparvovirus* in Alpacas

**DOI:** 10.3390/v11080701

**Published:** 2019-07-31

**Authors:** Deepak Kumar, Suman Chaudhary, Nanyan Lu, Michael Duff, Mathew Heffel, Caroline A. McKinney, Daniela Bedenice, Douglas Marthaler

**Affiliations:** 1Kansas State Veterinary Diagnostic Laboratory, College of Veterinary Medicine, Kansas State University, Manhattan, KS 66506, USA; 2Department of Clinical Sciences, Cummings School of Veterinary Medicine at Tufts University, 200 Westboro Road, North Grafton, MA 01536, USA

**Keywords:** alpaca, virus, *Bocaparvovirus*, genome, next-generation sequencing, metagenomics

## Abstract

Viruses belonging to the genus *Bocaparvovirus*
*(BoV*) are a genetically diverse group of DNA viruses known to cause respiratory, enteric, and neurological diseases in animals, including humans. An intestinal sample from an alpaca (*Vicugna pacos*) herd with reoccurring diarrhea and respiratory disease was submitted for next-generation sequencing, revealing the presence of a BoV strain. The alpaca BoV strain (AlBoV) had a 58.58% whole genome nucleotide percent identity to a camel BoV from Dubai, belonging to a tentative ungulate BoV 8 species (UBoV8). Recombination events were lacking with other UBoV strains. The AlBoV genome was comprised of the NS1, NP1, and VP1 proteins. The NS1 protein had the highest amino acid percent identity range (57.89–67.85%) to the members of UBoV8, which was below the 85% cut-off set by the International Committee on Taxonomy of Viruses. The low NS1 amino acid identity suggests that AlBoV is a tentative new species. The whole genome, NS1, NP1, and VP1 phylogenetic trees illustrated distinct branching of AlBoV, sharing a common ancestor with UBoV8. Walker loop and Phospholipase A2 (PLA2) motifs that are vital for virus infectivity were identified in NS1 and VP1 proteins, respectively. Our study reports a novel BoV strain in an alpaca intestinal sample and highlights the need for additional BoV research.

Bocaparvoviruses (BoVs) belong to the genus *Bocaparvovirus* and are emerging pathogens of the Parvoviridae family. BoVs are nonenveloped, single-stranded DNA viruses with an icosahedral symmetry and were originally named according to their first identified members, bovine parvovirus (BPV) and minute virus of canine (MVC) [1]. In the past few years, novel BoVs have been identified in a variety of animals, including bats [2], camels [3], gorillas [4], marmots [5], pigs [6], and rodents [7]. BoVs are comprised of 21 species, including carnivore BoV 1–6, chiropteran BoV 1–4, lagomorph BoV 1, pinniped BoV 1 and 2, primate BoV 1 and 2, and ungulate BoV (UBoV) 1–6. A few new UBoVs have been identified in dromedary camels (tentatively UBoV7 and UBoV8) [3] but have yet to be classified by the International Committee on Taxonomy of Viruses (ICTV). 

Initially, the classification of parvoviruses required the isolation of the virus; however, reporting of the viral sequence containing all the non-structural and structural coding regions is now acceptable provided the genomic, serological, or biological data supports infectious etiology [8]. Most of the members of the *Bocaparvovirus* genus have been identified using molecular methods and lack isolation in cell culture [3,4,9]. Human BoVs cause severe respiratory and gastrointestinal infections in young children [10]. Bovine parvovirus (BPV) causes gastrointestinal and respiratory symptoms, reproductive failure, and conjunctivitis in cattle worldwide [11]. Another important member of the BoV genus, canine minute virus (MVC), causes sub-clinical disease and fetal infections often leading to neonatal respiratory disease or abortions [12]. However, Koch’s postulates have yet to be fulfilled to link newly emerging BoVs with the clinical disease in animals [1,3,5].

Alpaca (*Vicugna pacos*) are domesticated members of the new world camelids closely related to llama (*Lama glama*), guanaco (*Lama guanicoe*), and vicuna (*Vicugna vicugna*) [13]. Over the past couple of decades, alpacas have gained significant popularity as pets, show animals, and fiber animals in the United States, with a total of 264,587 alpacas registered in the US as of May 2019 [14]. A variety of viruses have been identified in alpacas, including bovine viral diarrhea virus, coronavirus, adenovirus, equine viral arteritis virus, rotavirus, rabies, bluetongue virus, foot-and-mouth disease virus, bovine respiratory syncytial virus, influenza A virus, bovine papillomavirus, vesicular stomatitis virus, parainfluenza-3 virus, West Nile virus, and equine herpesvirus [12,15,16,17,18,19,20]. However, BoVs have yet to be reported in alpacas. 

An alpaca farm in the mid-eastern United States reported recurrent diarrhea and respiratory failure in young alpacas, with a case fatality rate up to 100%. In 2017, an alpha coronavirus was identified as causing clinical disease in two animals, and vaccination was subsequently attempted. However, diarrhea and respiratory distress continued to occur in juvenile animals despite increased biosecurity measures and supportive herd management. In 2018, an intestinal sample from a deceased alpaca was submitted to Kansas State University Veterinary Diagnostic Laboratory for metagenomic next-generation sequencing (NGS) to further evaluate potential causes of disease. The intestinal sample was processed, extracted, and sequenced using previously described methods [21,22]. The raw data was analyzed using a custom bioinformatic pipeline [23]. Reads were trimmed, and the adapter/index sequences were removed using Trimmomatic [24], Sickle [25], and Scythe [26]. 

A total of 334,052 cleaned reads were classified as eukaryotes (41%), bacteria (28%), viral (7%), and other organisms (4%) by Kraken software, which applies a *k*-mer search strategy from a sequence database to taxonomically classify reads (Figure 1) [27]. Kraken revealed a majority of the viral reads (22,170) as BoV (77%); bacteriophages (14%); and miscellaneous viruses composed of retroviruses, bacteria viruses, and unclassified viruses (9%). Reads lacking classification (no hits, *n* = 67,604) and identified as viral reads (*n* = 22,170) were de novo assembled into contigs and BLAST (Basic Local Alignment Search Tool) against the National Center for Biotechnology Information (NCBI) database, identifying a contig with a 58.58% nucleotide percent identity to a camel BoV from Dubai (KY640435). A full-length genome (5155 nucleotides) of an alpaca BoV (AlBoV, GenBank number MK014742) had an average read coverage of 2440X. AlBoV was aligned with the 108 complete UBoV genomes from GenBank using Multiple Alignment using Fast Fourier Transform (MAFFT) [28] in Geneious Prime [29]. AlBoV shared a 57.77–58.58% whole genome nucleotide identity to the UBoV8 strains (Table 1). Recombination events were not detected in the UBoV alignment using RDP4 software [30], although single-stranded DNA viruses such as parvoviruses possess a mutation rate similar to single-stranded RNA viruses and a higher mutation rate than double-stranded DNA viruses [31].

AlBoV contained three open reading frames (ORFs), NS1, NP1, and VP1/VP2, which were 2154 bp (411 to 2564 bp), 507 bp (2701 to 3207 bp), and 1395bp (3194 to 4588 bp), respectively. ICTV indicates a new parvovirus species should have less than 85% amino acid identity of the NS1 protein with other parvovirus species. The AlBoV identified in the present study shared the highest NS1 amino acid percent identity (57.89–67.85%) with camel BoVs in UBoV8 (Table 1) and represents a tentative new BoV species, UBoV9. The ancillary protein NP1, which is a unique feature of BoVs, is known to influence RNA processing events by suppressing internal polyadenylation and splicing of an upstream intron [32]. Unlike some of the other BoV sequences, the coding region of NP1 of AlBoV did not overlap with the C-terminal region of NS1. Interestingly, VP1/VP2 gene of AlBoV was the shortest among the identified UBoVs in the GenBank (Figure 2). Frameshift mutation were lacking in the AlBoV VP1/VP2 sequence, and the nucleotide sequence after the VP1/VP2 stop codon varied among the 108 complete UBoV sequences in the GenBank.

To study the phylogenetic relationship between AlBoV and other UBoVs, whole genome, NS1, NP1, and VP1 phylogenetic trees were created using a maximum likelihood method (phyML), using 500 bootstrap replicates in Geneious Prime. The trees were curated in FigTree (available from http://tree.bio.ed.ac.uk/software/figtree/) and Adobe Illustrator CS6 (Adobe Systems Inc, San Jose, CA, USA). Whole genome and NS1 phylogenetic trees illustrated that AlBoV shared a common ancestor with the UBoV8 species from camels (Figure 3). All eight species of UBoV (1–8) illustrated clear grouping in phylogenetic trees, which was observed in NP1 and VP1 phylogenetic trees as well. 

To investigate and identify the presence of virulence attributes, AlBoV was screened for the ATP or GTP-binding Walker loop motif (GPASTGKT) and Phospholipase A2 (PLA2) motif with the calcium-binding loop and phospholipase catalytic residues; GPASTGKT and PLA2 were found in the NS1 and N-terminal of VP1 proteins, respectively (Figure 4). These protein motifs are conserved and are required for parvovirus infectivity. Phospholipase A2 activity, with the calcium-binding loop and phospholipase catalytic residues, is critical for efficient transfer of the viral genome from the late endosomes/lysosomes to the nucleus for the initiation of replication, and hence is considered essential for virus infectivity [33]. Mutations of critical amino acid residues in the VP1 protein of human parvovirus B19 induces a strong reduction in phospholipase A2 activity and virus infectivity [34]. Considering their vital role in parvovirus infectivity, PLA2 inhibitors are also targeted for antiviral drugs against parvovirus-associated diseases. The presence of the Walker loop and Phospholipase A2 motifs suggests that the newly identified alpaca BoV possesses the virulence determinants necessary to cause disease.

A new species of UBoV was identified in an alpaca intestinal sample. Bocaparvoviruses cause respiratory and gastrointestinal infections in humans, bovines, and other animals. However, the causation of clinical disease in this alpaca farm is unclear. Establishing an association between the presence of BoV and clinical disease would require comprehensive PCR testing of the alpaca farms. Moreover, Koch’s postulates is required to establish a virus–disease association. A causal association between the presence of BoV and clinical disease is often difficult due to prolonged viral shedding by the host after infection, high prevalence of BoV infection, and high rate of co-infections [35]. Nevertheless, the discovery of a new BoV will help in developing new PCR diagnostics to determine the prevalence of BoV in alpaca herds and also to develop vaccines to prevent clinical disease. Given the high mutation rate of BoVs and increasing domestication of alpacas, identification of a new BoV in alpaca presents a true risk of cross-species transmission to other mammals.

## Figures and Tables

**Figure 1 viruses-11-00701-f001:**
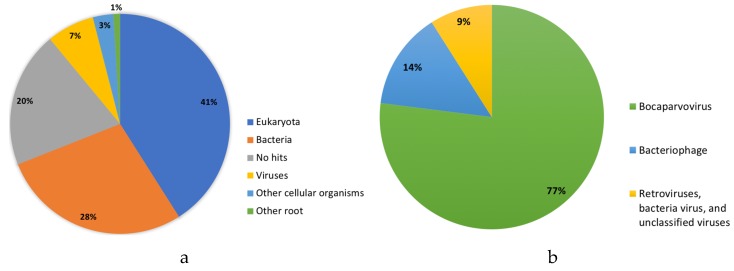
Metagenomic (**a**) and virome (**b**) results from the alpaca intestinal sample.

**Figure 2 viruses-11-00701-f002:**
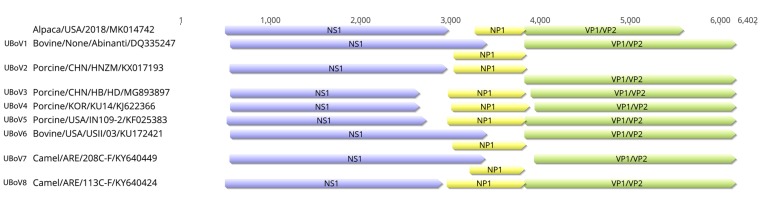
Genome organization of AlBoV compared to UBoV1–8. The purple, yellow, and green boxes indicate the NS1, NP1, and VP1/VP2 open reading frames (ORFs), respectively.

**Figure 3 viruses-11-00701-f003:**
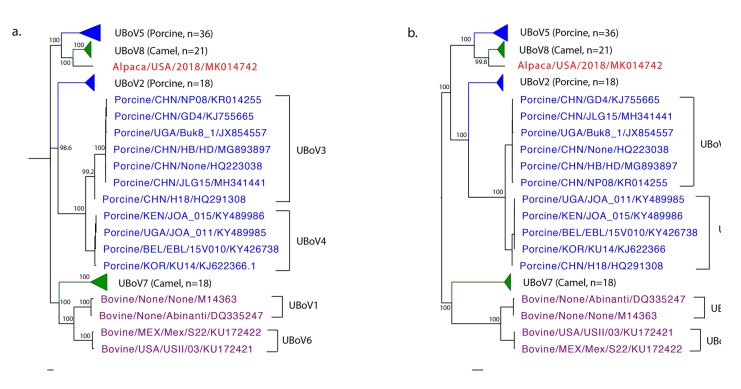
Mid-root Maximum Likelihood phylogenetic trees of AlBoV and other UBoV strains in the GenBank. (**a**) Whole genome phylogenetic tree. (**b**) Phylogenetic tree of NS1 amino acid sequences. Bootstrap values are indicated for major nodes. AlBoV is represented in red while the porcine, camel, and bovine UBoV strains are represented in blue, green, and purple, respectively.

**Figure 4 viruses-11-00701-f004:**
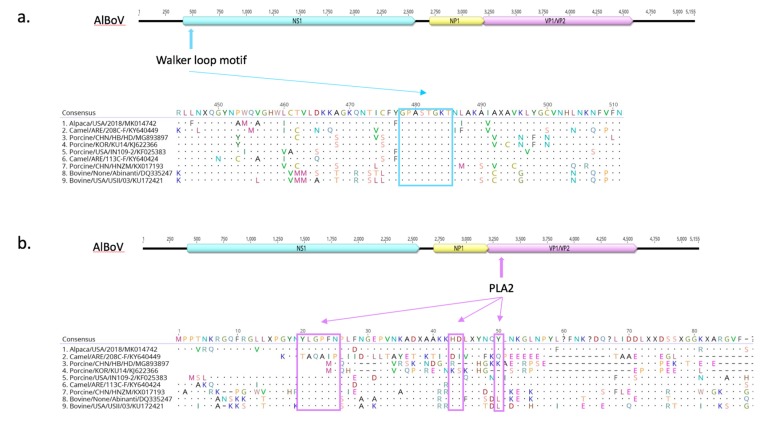
Detection of ATP and GTP-binding Walker-loop motif in NS1 protein (**a**) and Phospholipase A2 (PLA2) motif in N-terminal of VP1 protein (**b**). Dots indicate residues matching the consensus sequence and dashes represent gaps in the alignment.

**Table 1 viruses-11-00701-t001:** Nucleotide (nt) and amino acid (aa) percent identities of NS1, NP1, and VP1 genes of the alpaca bocaparvovirus (AlBoV) compared to the other ungulate BoV (UBoV) sequences (*n* = 108).

UBoV Virus	Species	Whole Genome (nt)	NS1		NP1		VP1	
			nt	aa	nt	aa	nt	aa
UBoV1 (*n* = 2)	Bovine	36.22–38.98	33.32–37.70	26.75–30.08	35.14–35.42	40.44–42.62	36.51–36.72	28.49–31.45
UBoV2 (*n* = 18)	Porcine	37.20–39.61	43.40–49.64	36.18–40.34	34.58–36.94	31.89–37.84	33.42–34.79	27.03–28.65
UBoV3 (*n* = 6)	Porcine	33.73–36.17	45.64–45.83	36.23–36.61	34.13–34.49	35.96	25.96–28.63	20.10–20.83
UBoV4 (*n* = 5)	Porcine	35.74–36.07	45.66–45.95	37.06–37.66	34.89–35.61	31.52–31.87	27.20–27.64	21.31–21.59
UBoV5 (*n* = 36)	Porcine	44.37–47.32	52.83–59.39	46.58–52.8	37.59–40.40	34.78–40.53	39.02–40.54	30.76–35.81
UBoV6 (*n* = 2)	Bovine	38.58–38.62	37.60–38.01	29.64	33.02	40.56–40.88	36.17–36.32	29.45–29.88
UBoV7 (*n* = 18)	Camel	36.49–36.76	39.01–39.64	33.08–33.98	36.91–38.93	32.24–33.88	26.05–27.11	21.08–21.84
UBoV8 (*n* = 21)	Camel	57.77–58.58	67.64–69.26	57.89–67.85	46.99–52.47	53.51–56.76	45.05–46.36	42.15–43.60

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
