# Peer review of "Metagenomic Next-Generation Sequencing Reveal Presence of a Novel Ungulate *Bocaparvovirus* in Alpacas"

_viruses, 2019, doi:10.3390/v11080701_

Round 1

Reviewer 1 Report

This paper describing the sequencing and analysis of a new bocaparvovirus isolated from an alpaca provides a nice, concise description of the virus sequence, its annotation, and potential taxonomic classification. As detailed below, the main concerns are proper use of taxonomic names, and proper referencing of bioinformatic tools.

Comments: 

·     Line 13: The name “Bocaparvovirus” is not being used as a taxon name in this sentence and therefore should not be italicized. (As an alternative, the sentence could read “Viruses belonging to the genus Bocaparvovirus are a…” with “Bocaparvovirus” italicized.

·     Line 17: Since the ICTV has not recognized a species named Ungulate bocaparvovirus 8, then this should not be referred to as a species.

·     Line 39: Change “Initially, the classification of Parvoviridae required…” to “Initially, the classification of parvoviruses required…”

·     Line 41-42: should read “of the Bocaparvovirus genus”.

·     Line 67: The “custom bioinformatic pipeline” needs to be described or referenced.

·     Line 68: references needed for Trimmomatic, Sickle and Scythe.

·     Line 71: reference needed for Kraken.

·     Lines 73-76 should be rewritten to improve clarity.

·     Line 75: The sequence (GenBank accession number MK014742) needs to be publicly available on publication.

·     Line 77: References needed for MAFFT and Geneious.

·     Line 79: Reference needed for RDP4.

·     Rest of paper: Properly reference all bioinformatic tools.

·     Lines 89-90: A species name is never italicized. I would recommend not trying to name the species and simply end the sentence: “…and represents a tentative new BoV species.”.

·     Table 1: Remove the word “species” from the Column 1 header. Perhaps replace with “virus” or “isolate”.

·     Lines 118 – 126: The function of the motifs is described, but the paragraph fails to discuss whether or not they were identified in AlBoV and if the sequence contains the indicated variants. 

·     Line 122: “…for the initiation of replication…”.

·     Lines 133-137: This discussion of virus discovery via metagenomic sequencing vs. PCR appears to be unnecessary since this is not a point relevant to this paper.

Author Response

General:

This paper describing the sequencing and analysis of a new bocaparvovirus isolated from an alpaca provides a nice, concise description of the virus sequence, its annotation, and potential taxonomic classification. As detailed below, the main concerns are proper use of taxonomic names, and proper referencing of bioinformatic tools.

Response: We thank reviewer for the suggestions. As suggested, variations in the taxonomic names have been corrected, and all the software and bioinformatic tools are now referenced. 

Comments

Line 13: The name “Bocaparvovirus” is not being used as a taxon name in this sentence and therefore should not be italicized. (As an alternative, the sentence could read “Viruses belonging to the genus Bocaparvovirus are a…” with “Bocaparvovirus” italicized.

Response: Thank you for the suggestion, the sentence has been changed.

Line 17: Since the ICTV has not recognized a species named Ungulate bocaparvovirus 8, then this should not be referred to as a species.

Response: We have amended the sentence indicating “tentative” Ungulate BoV 8 and 9 species.

Line 39: Change “Initially, the classification of Parvoviridae required…” to “Initially, the classification of parvoviruses required…”

Response:Corrected as suggested. 

Line 41-42: should read “of the Bocaparvovirus genus”.

Response: Corrected as suggested.

Line 67: The “custom bioinformatic pipeline” needs to be described or referenced.

Response: Corrected (line70)

Line 68: references needed for Trimmomatic, Sickle and Scythe.

Response: Corrected (line 71)

Line 71: reference needed for Kraken.

Response: A reference has been added for Kraken (line 74)

Lines 73-76 should be rewritten to improve clarity.

Response: Thank you for the suggestion. We have restructured the sentences for clarification (line 80-82). 

Line 75: The sequence (GenBank accession number MK014742) needs to be publicly available on publication.

Response: The full-length sequence of AlBoV was submitted to the Genbank on October 2, 2018. Sequence will be publicly available when the manuscript is accepted for publication.

Line 77: References needed for MAFFT and Geneious.

Response: Corrected (line 85)

Line 79: Reference needed for RDP4.

Response: Corrected (line 87)

Rest of paper: Properly reference all bioinformatic tools.

Response: All bioinformatic software are now referenced. 

Lines 89-90: A species name is never italicized. I would recommend not trying to name the species and simply end the sentence: “…and represents a tentative new BoV species.”.

Response: Sentence corrected as suggested (line 101).

Table 1: Remove the word “species” from the Column 1 header. Perhaps replace with “virus” or “isolate”.

Response: Word “species” is now replaced by “virus”

Lines 118 – 126: The function of the motifs is described, but the paragraph fails to discuss whether or not they were identified in AlBoV and if the sequence contains the indicated variants. 

Response:We thank reviewer for the suggestion. As suggested, a brief statement describing the detection of conserved Walker-loop and Phospholipase A2 motifs in alpaca BoV sequence has been included at line 134-138. Both of these protein motifs are conserved and essentially required for Parvovirus infectivity. Sentence describing variants/mutations in PLA2 motif (line 141-143) was included to highlight the importance of PLA2 motif in Parvovirus infectivity.

Line 122: “…for the initiation of replication…”.

Response: Sentence corrected as per suggestion (line 140-141).

Lines 133-137: This discussion of virus discovery via metagenomic sequencing vs. PCR appears to be unnecessary since this is not a point relevant to this paper.

Response: We agree that the discussion is not relevant to the paper. Hence, the paragraph has been deleted from the manuscript.

Reviewer 2 Report

A short report of a parvovirus sequence from an alpaca that is recovered from an intestine of an alpaca in an outbreak of severe disease in a young alpacas, although there is no evidence that the virus is associated with the disease in the animals. Metagenomic sequencing shows that there were sequences from a bocavirus in the materials collected from the alpacas. Some number of sequences were assembled into a 5155 nucleotide genome.

1) There is no real information about the coverage of the bocavirus sequences - how many reads were assembled to create the genome, how confident is the consensus sequence?

2) While the sequence had the features of a parvovirus, the VP1/2 gene was shorter than in other similar viruses, leaving a lot of supposedly non-translated sequences - that should be checked carefully to see if there is a frameshift in the sequence; check the 3 frames after the stop codon to see if one matches the bocavirus ORFs.

3) The sequence analysis of the different functional motifs is OK, but does not add much to the analysis.

4) Were PCRs carried out to determine the prevalence/association of this virus with the disease in alpacas?

5) What is the likelihood that this virus is the cause of the severe (or any) disease in alpacas? This should be carefully reviewed; superficially it seems that this is unlikely to be the cause, based on the results from other related viruses.

Author Response

1) There is no real information about the coverage of the bocavirus sequences - how many reads were assembled to create the genome, how confident is the consensus sequence?

Response: We thank reviewer for the comment. The number of “no hit” and “viral” reads are now mentioned at Line 82 and 84. In addition, we have included the average read coverage (2440) in the manuscript as well (Line 84). 

2) While the sequence had the features of a parvovirus, the VP1/2 gene was shorter than in other similar viruses, leaving a lot of supposedly non-translated sequences - that should be checked carefully to see if there is a frameshift in the sequence; check the 3 frames after the stop codon to see if one matches the bocavirus ORFs. 

Response: We thank the reviewer for this comment. We re-checked all alignments and sequences. We did not find any frameshift mutation in the sequence. The non-translated region of VP1/2 do not match with other BoV sequences in the Genbank.

3) The sequence analysis of the different functional motifs is OK, but does not add much to the analysis. 

Response:We thank reviewer the comment. Presence of conserved Walker loop motif and Phospholipase A2 motif is required for Parvovirus infectivity. In particular, PLA2 activity is required by parvovirus for its release from endosome. Their detection in the alpaca BoV sequence suggest that the newly identified BoV possess the virulence determinants necessary for causing disease. In view of the reviewer’s comment, we have expanded the paragraph for clarification (Line 134-138 and line 144-146). 

4) Were PCRs carried out to determine the prevalence/association of this virus with the disease in alpacas?

Response: No, PCRs were not carried in this alpaca farm to determine prevalence. After we detected a new BoV, the farm was advised for the PCR testing of other diarrheic alpacas to determine association between the newly identified BoV and disease in alpacas. Nevertheless, it would be beneficial to conduct a comprehensive PCR surveillance of alpaca farms to estimate prevalence of this parvoviruses in alpaca.

5) What is the likelihood that this virus is the cause of the severe (or any) disease in alpacas? This should be carefully reviewed; superficially it seems that this is unlikely to be the cause, based on the results from other related viruses.

Response: We thank reviewer for the suggestion.Bocaparvovirus cause respiratory and gastrointestinal infections in humans, bovines and other animals. However, it is not clear that they cause clinical disease in alpacas. Establishing any association between the presence of BoV and clinical disease requires comprehensive PCR testing of alpacas suffering from diahhrea and fulfillment of Koch’s postulates. However, identification of the sequence of a new BoV aids in the development of new PCR based diagnostics to determine the true prevalence and also to develop vaccines to prevent clinical disease. As suggested we have amended the paragraph and included reviewers’ suggestions (Line 159-167).

Reviewer 3 Report

Kumar and colleagues identified an ungulate Bocaparvovirus (UBoV) in an alpaca intestinal sample. They used next-generation sequencing data to show that this alpaca BoV strain has below 85% NS1 protein identity to any known UBoV and thus qualifies as a new species.

While this is an interesting discovery, PCR amplification of this new UBoV from samples obtained from few other alpacas with reoccurring diarrhea will make the conclusions more convincing.

Minor Comments:

1.       Lane 71: Please explain Kraken briefly and include reference.

2.       The authors should provide references for all the software used in the analyses.

Author Response

Kumar and colleagues identified an ungulate Bocaparvovirus (UBoV) in an alpaca intestinal sample. They used next-generation sequencing data to show that this alpaca BoV strain has below 85% NS1 protein identity to any known UBoV and thus qualifies as a new species.

While this is an interesting discovery, PCR amplification of this new UBoV from samples obtained from few other alpacas with reoccurring diarrhea will make the conclusions more convincing.

Response:  We agree with the reviewer that the PCR testing of more alpaca fecal samples will provide a more conclusive evidence about the disease causation. Although, we do not have access to more alpaca samples at the moment, the farm has been advised for PCR testing of more diarrheic alpaca fecal samples.  

Minor Comments:

Lane 71: Please explain Kraken briefly and include reference.

Response: A brief description of Kraken software and database has been added (line 74-78).

The authors should provide references for all the software used in the analyses.

Response: All software and bioinformatic tools are now referenced.